# Defining the Role of Oral Pathway Inhibitors as Targeted Therapeutics in Arteriovenous Malformation Care

**DOI:** 10.3390/biomedicines12061289

**Published:** 2024-06-11

**Authors:** Ann Mansur, Ivan Radovanovic

**Affiliations:** 1Division of Neurosurgery, Department of Surgery, Faculty of Medicine, University of Toronto, Toronto, ON M5S 1A8, Canada; 2Department of Laboratory Medicine and Pathobiology, School of Graduate Studies, University of Toronto, Toronto, ON M5S 1A8, Canada; 3Division of Neurosurgery, Department of Surgery, Toronto Western Hospital, University Health Network, Toronto, ON M5T 2S8, Canada

**Keywords:** arteriovenous malformation, vascular malformation, targeted therapy, liquid biopsy, precision medicine, clinical trials

## Abstract

Arteriovenous malformations (AVMs) are vascular malformations that are prone to rupturing and can cause significant morbidity and mortality in relatively young patients. Conventional treatment options such as surgery and endovascular therapy often are insufficient for cure. There is a growing body of knowledge on the genetic and molecular underpinnings of AVM development and maintenance, making the future of precision medicine a real possibility for AVM management. Here, we review the pathophysiology of AVM development across various cell types, with a focus on current and potential druggable targets and their therapeutic potentials in both sporadic and familial AVM populations.

## 1. Introduction

Brain arteriovenous malformations (bAVMs) are high-flow vascular malformations that involve abnormal shunting of blood directly from an artery to a vein without an intervening capillary network. These fragile vessels are prone to bleeding and proliferation, causing intracranial hemorrhage, edema, seizures, and focal neurological deficits [1]. They predominantly affect young individuals and result in significant morbidity and mortality in over one-third of patients harbouring this disease [2]. Appropriate and timely interventions are crucial in mitigating these risks and require a thorough understanding of the pathobiology of bAVM development and propagation in order to target existing and novel therapies to these relevant factors.

There has been a tremendous growth in knowledge of the underlying molecular drivers of vascular malformation pathogenesis, which largely lie in a few well-known signaling cascades that maintain angiogenesis, endothelial cell proliferation and survival, and the overall integrity of the neurovascular unit [3,4]. These have been widely implicated in oncological diseases, where numerous oral inhibitors have progressed through regulatory approval for those indications. The safety and utility of applying these pathway-targeted therapies for vascular malformations were studied first for slow-flow malformations and are reviewed elsewhere [4,5,6]. Here, we review our more recent knowledge on (1) the causative mutations driving bAVM development and (2) the complex interactions with various cell types of the neurovascular unit and inflammatory cascade that coordinate the stability of the vasculature. Importantly, we place this discussion within the scope of their existing and potential targeted therapeutics for bAVM care and highlight the most relevant druggable targets to be explored in future clinical trials.

## 2. Neurovascular Unit

The neurovascular unit is a well-described entity consisting of the endothelial cell surrounding a vessel lumen; it is intimately associated with perivascular mural cells, neuronal cells, supporting cells such as astrocytes, and inflammatory microglia.

Here, we discuss the three major cell types of the neurovascular unit (endothelial cells, mural cells, and inflammatory cells), their role in bAVM pathology, and any existing or potential targeted therapeutics for these specific factors.

### 2.1. Endothelial Cell

The endothelial cell (EC) is at the heart of angiogenesis in both the normal and pathological state. Various growth factors, adhesion factors, transcription factors, and extracellular matrix proteins interact with the endothelial cell membrane to trigger various signaling cascades that are crucial for EC permeability, differentiation, survival, growth, and migration.

#### 2.1.1. VEGF/VEGFR

VEGF is the one of the primary regulators of angiogenesis in the EC. While the VEGF family consists of seven members [7], VEGFA is the most relevant growth factor in the maintenance of various EC functions in the human, from permeability to survival and proliferation [7,8]. VEGF molecules bind to their tyrosine kinase receptors, VEGFRs, with VEGFR1 being present on a variety of cell types and VEGFR2 primarily found on ECs, lymphatic cells, and EC progenitor cells [9]. When VEGFA binds to VEGFR2 on ECs, the receptor dimerizes and autophosphorylates, which then causes downstream activation of two major cellular signaling pathways involved in EC survival, angiogenesis, and proliferation: the PI3K-AKT-mTOR and the RAS-RAF-MEK-ERK pathways [7].

The role of VEGF in bAVM development and pathogenesis has been demonstrated in various preclinical and clinical studies. Elevated VEGF levels have been correlated with bAVM hemorrhage and lethality [10]. Targeting VEGF reverted the aberrant vessel density and dysplasia in a mouse model [11]. Upregulation of angiogenic factors such as VEGF are observed in human bAVM compared to healthy quiescent ECs [12]. Furthermore, Chen et al. identified the upregulation of three VEGF-related microRNAs (miR7-5p, miR-199a-5p, miR-200b-3p) in the blood of patients with bAVM compared to healthy controls [13].

VEGFA is stimulated by various factors in the bAVM microenvironment, such as hypoxia and mechanical stress. Hypoxia has been shown to activate hypoxia-inducible transcription factor (HIF1a), which interacts with VEGFA (alongside NOTCH1 and ANPT2) to regulate angiogenesis [14]. Mechanical stress from high-flow in AVMs triggers the expression of VEGF, which has been supported by (1) the upregulation of VEGF and its receptor VEGFR2 in fast-flow malformations compared to slow-flow malformations, and (2) the upregulation of VEGF mRNA in AVM cells subjected to cyclic mechanical stretching compared to healthy human cells [15]. The response of VEGF to these stimuli is to induce vascular remodeling; when this is heightened in bAVM, it results in phenotypic structural changes such as increased elastic fibres, hypertrophy of the smooth muscle layer to overcome the shear stress and its eventual loss of contractility [16], and dilation of the vascular lumen [17].

The role of VEGF has also been demonstrated in familial bAVM syndromes, including hereditary hemorrhagic telangiectasia (HHT). Patients with HHT primarily harbour a germline loss-of-function mutation in either *endoglin (ENG*) or *activitin receptor-like kinase 1 (ALK1*) in HHT1 and HHT2, respectively. Both encode for important receptors of the transforming growth factor B (TGFB) superfamily, which in turn modulates the expression of various angiogenic genes [18]. Indeed, patients with HHT harbour elevated levels of both TGFB and VEGFA in their plasma [19]. Briefly, *ENG* is expressed in ECs and, through binding of BMP 9/10, it induces the BMP/ALK1 cascade that altogether suppresses endothelial cell proliferation and angiogenic gene expression [18,20]. Hence, loss-of-function germline mutations, alongside a second somatic hit in these genes, lead to marked angiogenic disinhibition and aberrant vessel morphology [4,6,20]. The role of VEGF in this interaction has been demonstrated in *ALK1* knockout mice, where VEGF signaling is required to develop bAVMs and treatment with a VEGF inhibitor, bevacizumab, attenuated this angiogenesis [11,21].

The VEGFA-VEGFR2 complex is therefore an attractive target in bAVM populations, including sporadic and familial cases (Figure 1). As mentioned, bevacizumab (Avastin) is a monoclonal antibody that directly inhibits VEGF; it has been approved to treat malignancies as well as retinal neovascularization. Bevacizumab was provided off-label to three adult patients with palliative sporadic extracranial AVMs with successful improvement in lesion deformity and patient symptomatology [15]. Several case reports have been published on the use of bevacizumab for the treatment of radiation effects after radiosurgery for bAVMs [22,23]. They observed a marked improvement in symptoms and a resolution of perilesional edema just one year after radiosurgery (possibly suggesting the enhanced antiangiogenic effect afforded by VEGF inhibition) [22]. To date, there has only been one pilot clinical trial published on bevacizumab for sporadic bAVMs; while the trial aimed at recruiting ten participants, it was terminated after the enrolment of two participants due to insufficient funding. These two patients with large palliative bAVMs experienced temporary reductions in serum VEGF levels, with no corresponding improvements in bAVM volume radiologically at either 26 or 52 weeks [24]. Unfortunately, this trial was underpowered to make any meaningful conclusions on the efficacy of bevacizumab on lesion morphology and the clinical phenotype in patients with sporadic bAVM. More trials are needed to investigate the role of VEGF inhibitors on both sporadic extracranial and intracranial populations.

A greater body of knowledge has been amassed on the safety and efficacy of bevacizumab for patients with HHT (Table 1). Various observational studies have been conducted on patients with HHT, demonstrating the efficacy of bevacizumab, primarily for recurrent epistaxis, as well as body AVM morphology [25,26,27,28,29,30,31] and severe gastrointestinal bleeding [32]. A multi-center observational study (InHIBIT-BLEED) on 238 patients with HHT receiving bevacizumab observed reductions in the frequency of red blood cell and iron transfusions of 82% and 70%, respectively, after 6 months [33]. They also found improvements in mean hemoglobin and reduced epistaxis severity after 1 year of treatment.

While there are no specific antiangiogenic drugs approved by federal drug authorities for HHT, the amounting evidence on bevacizumab has led to The Second International HHT Guidelines recommending the off-label use of systemic bevacizumab as standard-of-care, specifically for patients with significant bleeding that is refractory to maximal IV iron repletion [34]. Bevacizumab induction therapy starts with four to six biweekly infusions at 5 mg/kg; those who observe significant improvements in epistaxis and dependence on hematological supports are then bridged to maintenance treatment, which most often is recommended as continuous infusions scheduled every 4–12 weeks [33,35,36]. The most common side effects in patients with HHT include hypertension, proteinuria, and pain, all of which are generally well managed conservatively or with medical management [36]. Data from observational studies highlight that (1) the discontinuation of bevacizumab can result in recurrence of bleeding at the pre-treatment rate [19,36], while (2) chronic maintenance therapy (greater than 2 years) does not increase the risk of adverse events (although its long-term effects over decades are unknown) [35,36]; hence, those deriving therapeutic benefit can be maintained on the drug if they are tolerating it well.

In patients with severe HHT that progress on bevacizumab or have contraindications to it, treatment with pazopanib (Votrient) might be considered (Figure 1, Table 1). Pazopanib is an orally administered inhibitor of the VEGF/PDGF/c-kit receptors that is approved for advanced renal cell carcinomas and soft-tissue sarcomas. Preclinical studies demonstrated that treatment with a drug that mimics pazopanib prevents the development of AVMs in the *Acvrl1* mouse model [50]. Next, a phase I dose escalation study in seven patients with HHT showed significant improvements in epistaxis, hemoglobin, and quality of life in almost all patients, with no serious adverse events [34]. A more recent observational study on 13 patients with HHT and severe bleeding and transfusion dependence showed marked clinical improvements at low doses (25–300 mg daily) over 22 months, including in transfusion independence, in all patients after 12 months of treatment [48]. The most common side effects of pazopanib are hypertension, fatigue, and lymphocytopenia [48], with these being dose-dependent. The results will surely need to be corroborated by larger clinical trial data (US phase II/III trial NCT03850964 is underway) in order to comment on its adoption into clinical practice, and as such, patients with HHT refractory to maximal medical therapy might be considered ideal candidates for this line of inquiry.

#### 2.1.2. RAS-RAF-MEK-ERK Cascade

Stimulation of receptor tyrosine kinases on the EC membrane triggers dynamic hierarchical signaling networks in the EC. The initial inquiry into the pathogenesis of autosomal dominant familial syndromes such as capillary malformation–AVM (CM-AVM) highlighted the effect to these germline mutations on the abnormal regulation of the RAS-RAF-MEK-ERK cellular signaling cascade.

CM-AVM can be broken down into two distinct groups: CM-AVM type 1 and CM-AVM type 2. The former is characterized by a germline loss-of-function mutation in *RASA1*, which encodes a protein activator inhibiting the RAS/MAPK/ERK signaling pathway. The latter is caused by a mutation in the transmembrane receptor EHB4, which also normally works to inhibit the same signaling pathway [51]. Hence, these mutations lead to constitutive activation of the RAS/MAPK/ERK pathway, which governs cellular angiogenesis, proliferation, and motility. A reasonable target, hence, for these diseases is any effector of the pathway: from direct KRAS inhibitors to MEK inhibitors. To date, there has been just one case report of a 16-year-old female patient with severe CM-AVM type 2 that was treated with trametinib, an oral allosteric MEK1/MEK2 inhibitor, and experienced improvement in cardiac shunting after 10 months of therapy with stability of mild–moderate cutaneous adverse events [47].

The relevance of this pathway was later solidified in studies focusing instead on sporadic bAVMs, which account for around 90% of all bAVMs. Next-generation sequencing studies found that 55–63% of sporadic bAVMs harbour activating *KRAS* mutations [52,53], which went up to 90% with deeper sequencing techniques [54]. A minority of bAVM patients harbour activating *BRAF* or *RIT1* mutations and those with extracranial AVMs harbour mutations primarily in the *MAP2K1* gene [55]. Altogether, these mutations converge on the same signaling network. In vitro and immunohistochemistry studies demonstrate the resultant increased ERK phosphorylation, expression of angiogenic genes, enhanced migratory behaviour, and abnormal cell barrier phenotypes with a leaky phenotype [52,53,54,55,56]. Their activation also triggers the overexpression of genes related to angiogenesis, cell adhesion, cell migration, and tubular formation [55]. Preclinical models confirmed that KRAS pathway mutations are sufficient to induce bAVMs with similar resultant abnormal phenotypes, including increased cell size, sprouting, presence of shunts, and hemorrhage [55,56,57,58]. The application of a MEK inhibitor reverted these molecular and morphological phenotypes in both in vitro and preclinical in vivo work, respectively [52,54,56,57].

Given the amounting evidence that KRAS pathway mutations are necessary and sufficient to cause bAVMs in almost all sporadic cases and that the aberrant phenotype is reverted by MEK inhibition, the natural next step was to assess its safety and efficacy in patients with AVMs (Figure 1). Isolated case reports illustrate the safe application of the MEK inhibitor trametinib in two pediatric patients with palliative AVMs of the chest wall. Both patients experienced a reduction in the volume, redness, and deformity of their lesion, with an associated reduction in blood supply to the AVM [40,46]. One of these patients also had a spinal extension of their AVM that responded to trametinib with reduced shunting, which was the first suggestion that trametinib might also afford a clinical benefit to AVMs of the central nervous system [41]. Of note is that one patient who intermittently stopped trametinib therapy did experience a resurgence of symptoms during the drug holiday, which is not uncommon in the oncological experience with these targeted therapies [41].

Currently, two American and one European phase II trials assessing the safety and efficacy of the MEK inhibitors trametinib and cobimetinib in the treatment of palliative extracranial AVMs in pediatric and adult populations are underway (NCT05125471, NCT04258046, EudraCT: 2019-003573-26) (Table 1). Lastly, a more recent prospective pilot phase II trial assessing the efficacy of trametinib in improving the AVM angioarchitecture pre-operatively in adult patients with surgical AVMs of the central nervous system and body is now actively recruiting participants (NCT06098872). Results from these trials will aid in our understanding of how best to target this pathway for various AVM subpopulations. Studies with upstream inhibitors such as direct KRAS G12C and G12D inhibitors are emerging in the oncological population, alongside more indirect therapies such as targeting the KRAS protein degradation [59,60,61]. These complex strategies to inhibit the KRAS signaling pathway will also require validation in the bAVM population.

#### 2.1.3. PI3K/AKT/mTOR Cascade

The PI3K/AKT/mTOR cascade is another significant signal transduction axis relevant for angiogenesis, cell growth, and motility, and it has been the main target of many systemic therapeutics in slow-flow malformations [4]. Their effects on high-flow malformations are less robust. Specifically, reports on the use of sirolimus (rapamycin) as an mTOR inhibitor for KRAS-mutant AVMs did not yield a positive clinical response; these patients instead demonstrated improvements in their symptomatology and AVM morphology when their systemic therapy was switched to trametinib [40,41]. Case series including patients with AVMs alongside other complex vascular anomalies showed poorer response in the few patients harbouring AVMs compared to slow-flow malformations [42]; the mutational burden was not described for these patients but given that they were sporadic AVMs, it is most likely that they harbour mutations predominantly in the KRAS signaling pathway. Lastly, a study on 10 patients with extracranial AVMs treated with sirolimus also demonstrated its poor efficacy in this population [43]. This can be explained mechanistically through the molecular findings that (1) endothelial cell-enriched (CD31+) cell cultures of KRAS-mutant AVM tissue showed increased phosphorylation of ERK1/2, but not p38 or AKT [52], and (2) the inhibition of the MAPK-ERK pathway abrogated the angiogenic gene signature but inhibition of the P13K pathway did not [52].

Instead, PI3K/AKT/mTOR inhibition may play a more important role in the management of HHT-related AVMs. HHT has been characterized not only by increased pro-angiogenic signaling through VEGF/VEGFR2 upon ALK1 silencing (as discussed previously), but also by the overactivation of the PI3K/AKT/mTOR pathway. Mouse models of HHT revealed increased downstream ribosomal protein S6 phosphorylation in retinal ECs [62]. Application of the PI3K inhibitor wortmannin partially reduced the retinal AVM number in this model [62]. Similarly, mTOR inhibitors have been shown to activate SMAD1/5/8 signaling in cell assays [63] and preclinical models [63,64]. Combining the mTOR inhibitor sirolimus with the VEGFR2 inhibitor nintedanib had a synergistic effect on blocking and reverting AVM morphology and improved clinical pathology such as anemia, CBC markers, and bleeding in two HHT mouse models [65]. The application of low-dose tacrolimus (blood trough level of 1.5–2.5 ng/mL) in isolated patients with HHT revealed that it was well tolerated and resulted in improvements in epistaxis [44] and in cutaneous and gastrointestinal telangiectasias [44]. An open-label pilot study assessing the safety and efficacy of oral sirolimus for patients with HHT experiencing moderate-to-severe epistaxis is currently underway and aims to enroll 10 patients and follow them with serial labs and clinical assessments of epistaxis and plasma biomarkers over the course of 9 months (www.clinicaltrials.gov, accessed on 1 June 2024, NCT05269849).

#### 2.1.4. TGFB Signaling Cascade

Lastly, the TGFB signaling cascade is a highly conserved pathway that comprises various ligands, including TGFBs, BMPs, and activins, that are secreted in latent forms and activated by cleavage with extracellular proteases. These activated ligands bind to one of five type 2 tyrosine kinase receptors, which in turn phosphorylate type 1 receptors to form a complex that regulates SMADs within the cell [66]. Finally, activated SMADs form a complex with SMAD4 and translocate to the nucleus to regulate the transcription of various angiogenic genes [67]. As previously mentioned, signaling can occur through SMAD-independent pathways such as the MAPK and PI3K signaling cascades.

TGFB signaling has been shown to be involved in cerebral sprouting angiogenesis, maintenance of EC stability, and maturation of the blood–brain barrier (BBB), with its effects being exerted through complex interactions involving the EC, pericytes, and astrocytes [66]. Briefly, activated integrins have been found to lie on a concentration gradient from the ventral to dorsal regions of the brain, with a stabilization of vessels observed in ventral regions and a highly sprouting phenotype observed in dorsal regions [68]. BMP/ALK1 signaling has been shown to inhibit EC proliferation and enhance recruitment of pericytes to stabilize a mature and quiescent vascular network [69]. Its signaling, alongside the NOTCH pathway, upregulates the expression of N-cadherin to maintain the BBB [70], and aberrant BMP signaling in neural stem cells results in abnormal astrocyte-EC interaction, increased VEGF expression, and formation of AVMs [71].

Decreased TGFB signaling has been observed in patients with bAVMs and its role in the pathogenesis of HHT is well described and has previously been elaborated on. The pharmacological inhibition of this pathway for patients with HHT has been demonstrated by targeting the PI3K pathway downstream, with a reduction in ALK1-induced vascular proliferation observed in vivo [66,72].

### 2.2. Mural Cells

Mural cells collectively refer to a heterogenous group of cells, including smooth-muscle cells (larger) and pericytes (smaller), which play an integral role in maintaining the structural integrity of the neurovascular unit and the blood–brain barrier [73]. Structurally, bAVMs are characterized by a weakened vessel wall with dilatations and arteriovenous shunting [17,21,74] that, at the histological level, correspond to hypertrophy of the muscular layer and increased elastic fibres with associated loss of collagen type 3 of vascular smooth muscle cells (vSMCs) [75]. Abnormalities in mural cell morphology and interactions with other cell types in bAVM have been described in relation to hemodynamic forces with resultant abnormal cellular pathway signaling, as described below [3].

Pericytes account for 80–90% of the cerebral vascular wall [76] and contribute to the vessel wall stability by inhibiting the degradative metalloproteinases (MMPs) and inducing endothelial tight junction complexes [73]. Mouse models demonstrate the increased diameter of bAVM vessels with a loss of surrounding pericytes [74]. This in turn leads to increased permeability to surrounding circulating factors and mechanical weakness of the vascular wall with progressive dilatation and arteriovenous shunting [77,78]. These findings were corroborated by human immunofluorescence models that found reductions in pericyte number and density in bAVM tissue compared to temporal lobe tissue, reduced pericyte coverage in ruptured bAVMs, and a correlation between reduced pericyte number with microhemorrhage severity and blood flow in the bAVM nidus [79]. Altogether, the evidence speaks to the involvement of mural cells in determining the fragility of the vascular bed and resultant neurovascular coupling [80]. The complex cellular interactions that govern cellular integrity can be summarized primarily by the PDGFB/PDGFR and ANGPT/TIE signaling pathways, both of which are involved in bAVM pathology.

#### 2.2.1. PDGFB/PDGFR

The PDGF family consists of four types of mitogens (PDGFA, PDGFB, PDGFC, and PDGFD) that bind to either the PDGF alpha or beta receptors on various mesenchymal cells [81]. They are involved in a variety of cellular functions ranging from cell growth, differentiation, and wound healing to angiogenesis. PDGFB is one of the most widely studied mitogens in the family due to its pro-angiogenic effects. The ligand/receptor complex activates the MAPK/ERK and P13K/AKT pathways, which in turn increases the proliferation and migration of mesenchymal cells [82] and ultimately regulates pericyte recruitment [83,84]. The PDGFB-mediated activation of these pathways is also regulated upstream by EphrinB2 on the vSMC surface, with its ablation enhancing MAPK activation [85]. Knockout of *PDGFB* or its receptor in a mouse model confirmed the loss of pericytes with associated cerebral microvascular hemorrhage [85,86] and high embryonic lethality [87]. PDGFB expression was also reduced in bAVM lesions in an Alk-1 deficient mouse model [74], establishing some connection between the pathways. Furthermore, excessive VEGFA signaling in ECs has been shown to affect the transcription of PDGF-B, with decreased PDGF-B observed in vitro and ex-vivo models of angiogenesis [88]. Similarly, bAVMs have been shown to harbour elevated levels of the TGFB1 protein, and mRNA, which binds to ALK5 through Smad2/3, also affecting PDGFB transcription [89]. Hence, PDGF-B/PRGFR signaling is crucial for mural cell differentiation and plasticity in bAVM pathogenesis [90] and it is intimately tied with other key regulators of bAVM angiogenesis.

The regulation of PDGFB expression in bAVM preclinical studies was first studied using a lentiviral vector-mediated gene transfer which resulted in an increase in mural cell coverage and a reduction in both dysplastic vessels and their associated hemorrhage [91]. Direct PDGFR inhibitors such as sunitinib and ponatinib have been studied in oncology to significantly inhibit cell growth, decrease MMP expression, and enhance the activity of other chemotherapeutics [50]. Their utility in AVMs has yet to be explored. Instead, more indirect mechanisms of targeting this pathway have been highlighted in both familial and sporadic cases. Thalidomide is a drug that used to be prescribed for morning sickness in pregnant women and while it was discontinued for its teratogenic effects in this population, the discovery of its antiangiogenic effects has caused its resurgence in vascular research (Figure 2). Thalidomide exerts its antiangiogenic effect by targeting various angiogenic growth factors such as VEGF, PDGF, and TGFB alongside MMPs in extracellular matrix remodeling [50,92,93,94]. Studies on peritoneal angiogenesis reveal that thalidomide exerts an inhibitory effect on several endothelial targets, including VEGFR2, VEGFR3, and STAT3/SP4 signaling [92]. It suppresses the expression of VEGFR in human umbilical vein ECs via the activation of the sphenolipid signaling pathway [86]. Thalidomide also decreases the secretion of VEGF, basic fibroblast growth factor (bFGF), and hepatocyte growth factor in bone marrow ECs of active multiple myeloma and Kaposi’s sarcoma cell lines [93]. Treatment with its analogue lenalidomide not only reduced VEGF and TGFB secretion but also affected the aberrant cell migration, cell adhesion, AKT phosphorylation, and capillary tube formation in human endothelial cell lines, speaking to its effect on multiple downstream signaling pathways [94].

In bAVM research specifically, thalidomide treatment reduced bAVM hemorrhage and vessel dysplasia in a mouse model [91]. In the setting of HHT, high-dose thalidomide treatment (100 μg/mL) inhibited normal vessel formation, whereas low-dose (25 μg/mL) treatment restored vessel wall weakness by increasing PDGF-B expression in ECs and stimulating mural cell activation [38]. Given the evidence of its effectiveness in preclinical models, thalidomide has made its clinical resurgence in various studies on vascular disorders in human populations. The safety and efficacy of thalidomide in the treatment of recurrent bleeding in the context of gastrointestinal vascular malformations and angiodysplasias has been studied in various observational studies and clinical trials, demonstrating its efficacy in reducing bleeding event rates and circulating VEGF levels, with mostly minor adverse events even in those with significant comorbidities [95,96,97,98]. Similarly, there are various reports highlighting the clinical efficacy of thalidomide on reducing bleeding events in patients with HHT [38,39]. More recently, Boon and colleagues reported on an observational case series of 18 adult patients with severe extracranial AVMS with functional impairments that were treated with thalidomide [37]. They found that the patients reported reductions in pain, alongside improvements in bleeding and ulceration. Of the twelve patients who stopped thalidomide due to clinical stability, four had eventual lesion recurrence within one year. Of the patients who were trialed on the higher dose (200 mg daily), 80% experienced grade 3 complications, while those who were maintained on the lower dose (50 mg daily) had no severe adverse events with similar clinical efficacy. These reports suggest that thalidomide might be a promising drug to further study in the bAVM population given its multi-faceted role in angiogenic control.

#### 2.2.2. ANGPT/TIE Signaling

The human ANGPT-TIE system consists of three ligands (ANGPT1, ANGPT2, and ANGPT4) that bind to either TIE1 or TIE2 receptors. ANGPT1 and ANGPT2 are the predominant ligands involved in human angiogenesis and vascular homeostasis; their functions are complex and uniquely context-dependent [99]. ANGPT1 is found primarily on mural cells and fibroblasts, whereas ANGPT2 is linked to ECs that are actively remodeling. ANGPT1 functions as a TIE2 receptor agonist, which, when activated, will stimulate multiple downstream signaling cascades to maintain EC survival and cell barrier. ANGPT2 negatively regulates ANGPT1, leading to enhanced angiogenesis and vascular permeability in the context of actively remodeling ECs. These remodeling cells are triggered by inflammatory cytokines, hypoxia, and by pro-angiogenic growth factors such as VEGF. Interestingly, in the absence of ANGPT1, ANGPT2 has been shown to be a weak TIE2 agonist, illustrating its context-dependent functional activity [100].

When devising targeted therapeutics for the ANGPT/TIE system in oncological models, considerations were made to each ligand, receptor, and their complex situation-dependent activity in order to achieve the desired clinical effect. Information from these investigations reveals fundamental principles that must be considered when repurposing drugs for neurovascular conditions. For example, the stabilizing effects of ANGPT1 are not all clinically beneficial given that if it stabilizes an already quiescent vasculature, it will maintain that quiescent state, but if there is active angiogenesis (as in the case of bAVM pathogenesis), then this active state might be maintained [99]. Hence, these conflicting results make ANGPT1 a non-ideal target for bAVM treatment. Instead, selective ANGPT2 inhibition has been shown to block angiogenesis, restore cellular junctions, and increase pericyte coverage in oncological models [101]; this is perhaps attributed to the overexpression of ANGPT2 in ECs that are undergoing vascular remodeling and are abnormally active in pathological states (Figure 2). Clinical trials on ANGPT2 inhibitors demonstrate significant reductions in tumour blood flow [99], but as monotherapies, they have proven less effective in preventing tumour progression in clinical trials than in experimental models [102].

The implication of the ANGPT/TIE system in VMs has been studied in the context of slow-flow capillary malformations, where a somatic missense mutation in *GNAQ* in ECs results in the overactivation of the PKC-NFkB-ANGPT2 pathway [103], with increased ANGPT2 expression observed in mouse xenograft and human capillary malformation studies [104]. Elevated ANPGT2 was also observed in mouse models of HHT [105] alongside patient tissue from a bAVM [105]. ANGPT2 traps, especially in conjunction with upstream angiogenic factor inhibition, might be an attractive therapeutic intervention to study in future preclinical bAVM models [105,106].

### 2.3. Inflammatory Cells

Inflammatory cells have an important role in the maintenance and propagation of bAVM angiogenesis, destruction of basement membranes, and recruitment of pericytes. There are a wide variety of inflammatory cells present in the various layers of the neurovascular unit, including macrophages, monocytes, neutrophils, and B and T lymphocytes. Macrophages/microglia make up the vast majority of the inflammatory cells observed in bAVM vascular beds; their presence does not correlate with rupture status, implying their inherent role in bAVM pathogenesis and not just a reaction to hemorrhage [107]. It is unclear which insults trigger the infiltration of inflammatory cells into the bAVM neurovascular unit; however, several factors allow for its abnormal and heightened inflammatory response [108]. Firstly, the presence of an already leaky perivascular unit physically enables the infiltration of various inflammatory cells and cytokines. Next, the infiltration of macrophages seems to be impaired in bAVM animal models; specifically, the recruitment of macrophages is seen later in bAVM development and is maintained with persistent infiltration and abnormal clearance [109]. Next, the presence of physical factors such as shear stress in bAVMs activates pro-inflammatory signaling in ECs through the autocrine signaling pathway with NF-KB, resulting in macrophage aggregation [110].

An increased macrophage burden subsequently (1) supports angiogenesis through increased VEGF secretion for binding to VEGFR on ECs and through direct influence on the ANGPT-TIE pathway [111]; (2) enhances recruitment of pericytes through upregulation of PDGFB [112]; and (3) enables aberrant vascular remodeling with the degradation of the basement membrane through various MMPs [108] and through the upregulation of autocrine feedback loops with the COX2-PGE2-NFKB pathway [113].

Given that both ruptured and unruptured bAVMs are linked to inflammatory processes, researchers logically aimed at targeting this process with anti-inflammatory drugs in preclinical models. One study showed that treatment with minocycline or pyrrolidine dithiocarbamate attenuated cerebral MMP-9 activity and VEGF-induced hemorrhage in a bAVM mouse model [99]. Similarly, the use of doxycycline lowered MMP-9 levels and vascular density in a mouse model [114]. The effect of doxycycline was later explored in 10 human bAVM patients who received either 100 mg of doxycycline or placebo twice a day for one week prior to AVM resection. While there was a trend towards a reduction in MMP-9 levels, this did not reach significance [115]. MMP inhibitors also largely failed in oncological trials and demonstrated severe adverse reactions, making them less favourable therapeutics to pursue at this time [49,116]. Similarly, the use of corticosteroids can be beneficial in select situations, such as for the management of radiation effects in bAVM management [2], but they are not intended for chronic use in AVM management or for patients with venous congestive myelopathy secondary to spinal AVMs [117].

## 3. Promising Drug Targets for bAVM Care: Conclusions and Future Directions

bAVM induction and maintenance is clearly reliant on an underlying mutational hit with a subsequent involvement of various pathological factors and signaling networks between the EC, mural cells, and the inflammatory network in order to destabilize vessel wall integrity. An approach to targeting the underlying pathobiology of this disease will require a more in-depth and exhaustive understanding of these intricate networks and how they specifically behave in both bAVM development and their response to the microenvironment. Based on our current body of knowledge, mutation in the genes of the KRAS signaling pathway have been shown to be necessary and sufficient to induce bAVMs, making this pathway an attractive target for future clinical studies and patient trials. Further knowledge is required on whether these drugs perform equally well in extracranial AVMs and those of the central nervous system, given the role of the microenvironment on AVM transcriptional regulation [118]. Further research is also needed to discern whether an approach using small-molecule inhibitors of this pathway or the degradation of their resultant proteins yields a better clinical and radiological response.

Next, targeting master regulators of angiogenesis, such as VEGFA, and those that govern mural cell function seems logical given their implications in bAVM pathology in both preclinical and clinical research. There are evidently numerous factors that support the dysfunctional neurovascular unit; hence, monotherapeutic approaches will likely not be the most clinically impactful. Instead, combinatorial approaches will help to address the redundancy in angiogenic drivers and their drug resistance while helping to reduce unintended adverse events. For example, the clinical effect of ANGPT2 blockade has been shown to be augmented when combined with inhibitors of VEGF, likely because VEGF activation is required for pathological ANGPT2 activity [102]. The use of drugs that have multiple desired targets, such as thalidomide and pazopanib, might also enable a multi-pronged approach when combined with KRAS pathway inhibitors. Future preclinical studies are required to understand which of these combinatorial drug approaches will best normalize the cerebral vasculature and which modes of drug delivery can target these pharmacological agents directly to the abnormal tissue to decrease off-target effects.

## Figures and Tables

**Figure 1 biomedicines-12-01289-f001:**
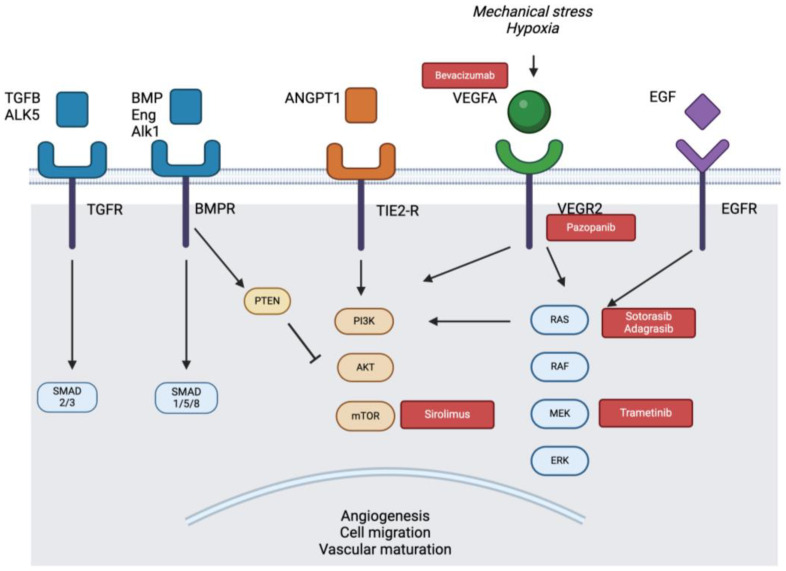
Schematic of relevant pathways and their targeted inhibitors (in red) for the EC. Stimuli trigger VEGFA to bind and autophosphorylate VEGFR2, triggering both the PI3k/AKT/mTOR and RAS/RAF/ERK pathways. Patients with HHT have germline mutations in the TGFB pathway that result in ALK1 activation by BMP9 and 10 ligands, which can trigger the TGFB canonical (SMAD 1/5/8) and non-canonical pathways (PI3K/AKT/mTOR). Patients with capillary malformation AVM syndrome harbour germline mutations in RASA1 and EPHB4, leading to the KRAS signaling pathway. Patients with sporadic AVMs harbour mostly *KRAS* mutations, alongside *BRAF*, *MAP2K1,* and *RIT1*. Activation of these pathways results in the expression of proteins important in angiogenesis, vascular migration, and maturity.

**Figure 2 biomedicines-12-01289-f002:**
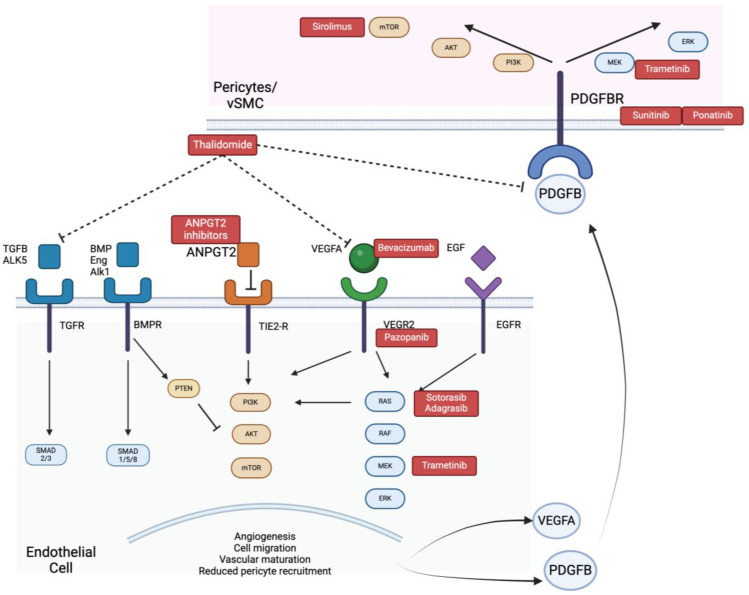
Schematic of interaction between activated ECs and pericytes/vascular smooth-muscle cells (vSMCs) and their targeted inhibitors in red. PDGFB secreted from the EC binds to its receptor on the mural cell, causing activation of both P13K/AKT/mTOR and RAS/RAF/ERK pathways. Activated ECs have greater ANGPT2 expression, which in the presence of ANGPT1 will inhibit TIE2 receptors, causing enhanced vascular permeability. Altogether, there is decreased pericyte recruitment, enhanced MMP activation, and reduced vascular stability in this interaction.

**Table 1 biomedicines-12-01289-t001:** Summary of observational studies and clinical trials on oral targeted therapies in humans with sporadic AVMs and familial AVM syndromes.

Drug	Target	Sporadic AVMs	Familial AVM Syndromes
Bevacizumab (Avastin)	VEGFA	*Observational studies:* ○ Improved lesion deformity and symptoms in extracranial severe AVMs [15] ○ Improved symptoms and perilesional edema after radiation [22,23] *Completed clinical trials:* ○ Two patients had reduced serum VEGF with no change in bAVM on imaging at 26 or 52 weeks [24] (USA, California, phase I, NCT02314377)	*Observational studies:* ○ Clinical improvement in recurrent epistaxis, GI bleeding, infusion dependency, CBC parameters, and body AVM morphology in HHT [25,26,27,28,29,30,31,32,33,34,35,36]
Thalidomide	VEGF, PDGF, TGFB	*Observational studies:* ○ Thalidomide resulted in improvements in symptomatology, bleeding, and ulceration in adult patients with palliative extracranial AVMs with a dose-dependent adverse event profile [37]	*Observational studies:* ○ Thalidomide has been shown to be effective in treating recurrent bleeding and lowering VEGF in patients with HHT [38,39] *Completed clinical trials:* ○ Low-dose thalidomide afforded significant improvements in epistaxis in 90.3% of patients with HHT and a complete cessation of bleeding in 9.7% of patients with HHT (Europe, Italy, phase II open lab, NCT01485224)
Sirolimus/ Tacrolimus	mTOR	*Observational studies:* ○ Observational studies on mTOR inhibitors for pediatric and adult patients with extracranial sporadic AVMs do not show a robust improvement in AVM symptomatology or morphology [40,41,42,43]	*Observational studies:* ○ Low-dose tacrolimus resulted in improvements in epistaxis and telangiectasias in isolated patients with HHT [44,45] *Ongoing clinical trials:* ○ Pilot phase II study on low-dose sirolimus for nosebleeds in HHT (Canada, Toronto, NCT05269849)
Trametinib/ Cobimetinib	MEK	*Observational studies:* ○ Case reports of pediatric patients with palliative extracranial AVMs illustrate improvement in symptomatology, AVM, and flow with daily trametinib [40,41,46] *Ongoing clinical trials:* ○ COBI-AVM phase II Study: daily cobimetinib for palliative extracranial AVMs in pediatric and adult populations (USA, Arkensas, NCT05125471) ○ Phase II trial on trametinib for palliative extracranial AVMs in pediatric and adult populations (USA, California, NCT04258046) ○ TRAMAV: European phase II trial on trametinib for palliative extracranial AVMs in pediatric and adult populations (Europe, Belgium, EudraCT: 2019-003573-26) ○ OZUHN-017: Canadian phase II pilot trial on trametinib in improving angioarchitecture in pre-surgical unruptured adult AVMs of the brain and body (Canada, Toronto, NCT06098872)	*Observational studies:* ○ Case report on pediatric patient with CM-AVM type 2 showed improvement in cardiac shunting and AVM stability after 10 months of daily trametinib [47]
Pazopanib (Votrient)			*Observational studies:* ○ In 13 patients with HHT, low-dose daily pazopanib resulted in improvements in bleeding, transfusion dependence, and CBC parameters in all patients after 12 months of treatment [48] *Completed clinical trials:* ○ Phase I dose escalation study in 7 patients with HHT showed improvements in epistaxis, hemoglobin, and QoL with no serious adverse events [34] *Ongoing clinical trials:* ○ US phase II/III trial (NCT03850964) by Cure HHT is assessing effect of pazopanib on epistaxis and anemia in HHT
Doxycycline	MMP-9	*Completed clinical trials:* ○ Placebo-controlled trial on doxycycline given twice a day for 1 week prior to surgery showed a non-significant reduction in MMP-9 levels [49]	

## Data Availability

Not applicable.

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
