# Peer review of "Defining the Role of Oral Pathway Inhibitors as Targeted Therapeutics in Arteriovenous Malformation Care"

_biomedicines, 2024, doi:10.3390/biomedicines12061289_

Round 1

Reviewer 1 Report

Comments and Suggestions for Authors

This review insightfully summarized bAVM research regarding signaling pathways and vascular units in bAVMs and focused on drugs relevant to bAVMs. It is suggested to add overall TGFβ signaling and the crosstalk of astrocytes with the vascular unit in bAVMs. Also, it will be valuable information if the authors can suggest examples of small molecules or drugs that have the potential to be used in bAVM treatment. Additional minor comments are listed below,

-       - Some abbreviations need to be unified. For example, VEGFR2 or VEGFR-2, TGF-B or TGFB. Some sentences miss the reference, for example, in line 99-101 “Hence, loss-of-function germline mutations… aberrant vessel morphology”, line 151-155 “Data from observational…adverse events”. Please check these throughout the manuscript.

      - The study by Park et al. (doi: 10.1002/ana.26059) can be added to the preclinical evidence to show that KRAS mutation is sufficient to induce bAVM development.

Author Response

We thank Reviewer 1 for his/her thoughtful review and comments. We revised our manuscript to include the many helpful suggestions with elaboration on TGFB signaling, a comment on its role in astrocyte-EC interactions, a summary table of the relevant oral drugs for AVM care, and more thorough citations throughout. We hope that these changes will improve the quality of our seminal review and that it will be appropriate for publication.

Reviewer 2 Report

Comments and Suggestions for Authors

See attached.

Author Response

We thank Reviewer 2 for his/her thoughtful review and comments. We revised our manuscript to include the many helpful suggestions with elaboration on mTOR inhibitors in sporadic and familial AVM populations, a summary table of the relevant oral drugs for AVM care, and more thorough citations throughout. We hope that these changes will improve the quality of our seminal review and that it will be appropriate for publication.
